# Subcellular Localization of Fad1p in *Saccharomyces cerevisiae*: A Choice at Post-Transcriptional Level?

**DOI:** 10.3390/life11090967

**Published:** 2021-09-14

**Authors:** Francesco Bruni, Teresa Anna Giancaspero, Mislav Oreb, Maria Tolomeo, Piero Leone, Eckhard Boles, Marina Roberti, Michele Caselle, Maria Barile

**Affiliations:** 1Department of Biosciences, Biotechnologies and Biopharmaceutics, University of Bari Aldo Moro, Via Orabona 4, 70125 Bari, Italy; francesco.bruni@uniba.it (F.B.); teresagiancaspero@virgilio.it (T.A.G.); maria.tolomeo@uniba.it (M.T.); pieroleone87@gmail.com (P.L.); marina.roberti@uniba.it (M.R.); 2Institute of Molecular Biosciences, Goethe-University Frankfurt, Max-von-Laue-Str. 9, 60438 Frankfurt am Main, Germany; m.oreb@bio.uni-frankfurt.de (M.O.); e.boles@bio.uni-frankfurt.de (E.B.); 3Physics Department, University of Turin and INFN, Via P. Giuria 1, 10125 Turin, Italy; michele.caselle@unito.it

**Keywords:** FAD synthase, *FAD1*, mitochondria localization, *Saccharomyces cerevisiae*, mRNA, mitochondrial localization motif

## Abstract

FAD synthase is the last enzyme in the pathway that converts riboflavin into FAD. In *Saccharomyces cerevisiae*, the gene encoding for FAD synthase is *FAD1*, from which a sole protein product (Fad1p) is expected to be generated. In this work, we showed that a natural Fad1p exists in yeast mitochondria and that, in its recombinant form, the protein is able, per se, to both enter mitochondria and to be destined to cytosol. Thus, we propose that *FAD1* generates two echoforms—that is, two identical proteins addressed to different subcellular compartments. To shed light on the mechanism underlying the subcellular destination of Fad1p, the 3′ region of *FAD1* mRNA was analyzed by 3′RACE experiments, which revealed the existence of (at least) two *FAD1* transcripts with different 3′UTRs, the short one being 128 bp and the long one being 759 bp. Bioinformatic analysis on these 3′UTRs allowed us to predict the existence of a *cis*-acting mitochondrial localization motif, present in both the transcripts and, presumably, involved in protein targeting based on the 3′UTR context. Here, we propose that the long *FAD1* transcript might be responsible for the generation of mitochondrial Fad1p echoform.

## 1. Introduction

Riboflavin (Rf or vitamin B2) deficiency in humans and experimental animals has been linked to several diseases, such as cancer, cardiovascular diseases, anemia, abnormal fetal development, different neuromuscular and neurological disorders, some of which are treatable with high doses of riboflavin. Among the latter, there are the multiple acyl-CoA dehydrogenase deficiency (MADD) and the Brown–Vialetto–Van Laere syndrome (BVVLS) (see [1] and refs therein). The role of this vitamin in cell metabolism depends on its conversion into flavin mononucleotide (FMN) and flavin adenine dinucleotide (FAD), which are the redox cofactors of a large number of dehydrogenases, reductases, and oxidases. Most of these flavoenzymes are compartmented in the cellular organelles, where they are involved in energy production and redox homeostasis as well as in different cellular regulatory events including protein folding, apoptosis, and chromatin remodeling [2,3,4]. The relevance of such processes merits further research aimed to better describe flavin homeostasis and flavoenzyme biogenesis, especially in those organisms that can be simple and suitable models for human pathologies. The conservation of the main biological processes within all eukaryotes, together with the possibility of simple and quick genetic manipulation, make the budding yeast, *Saccharomyces cerevisiae,* a suitable model to understand the molecular mechanisms underlying human diseases [5,6,7,8].

Yeasts, as well as other fungi, plants and bacteria, have the ability either to synthesize Rf de novo or to take it from outside, whereas mammals must obtain Rf from diet. The first eukaryotic gene coding for a cellular Rf transporter was identified in *S. cerevisiae* as the product of the *MCH5* gene [9]. More recently, three human Rf transporters have been cloned and characterized; they belong to a novel family of Rf transporters, namely, RFVT/SLC52, which exhibit no significant similarity to Mch5p [10,11]. Mutations in RFVTs have been recently linked to BVVLS, a neurodegenerative disorder characterized by cranial nerve deficits, bilateral sensorineural deafness, respiratory insufficiencies, and the degeneration of some spinal cord neurons [1,12].

Intracellular conversion of Rf to FAD is a ubiquitous pathway and occurs via the sequential actions of ATP:riboflavin 5’-phosphotransferase or riboflavin kinase (RFK, EC 2.7.1.26), which phosphorylates the vitamin into FMN, and of ATP:FMN adenylyl transferase or FAD synthase (FADS, EC 2.7.7.2) that adenylates FMN to FAD. Eukaryotes generally use two different enzymes for FAD production, whereas most prokaryotes depend on a single bifunctional enzyme [13,14]. The first eukaryotic genes encoding for RFK and FADS were identified in *S. cerevisiae* and named *FMN1* [15] and *FAD1* [16], respectively.

Fmn1p is a 24.5 kDa protein showing a sequence and structure similarity to the RKF-module of prokaryotic FADS and appears largely conserved through evolution. Immunoblotting analysis of subcellular fractions revealed that Fmn1p is localized in microsomes and in mitochondria [15]. Orthologs of *S. cerevisiae FMN1* have been cloned and the corresponding proteins purified from the yeast *Schizosaccharomyces pombe* and from *Homo sapiens*. The crystal structures of both these proteins have been solved, revealing a novel ATP and Rf-binding fold [17,18,19].

Fad1p, the sole known protein isoform generated by *S. cerevisiae FAD1* gene, is a 35.5-kDa soluble enzyme that is essential for yeast life, whose crystal structure has been solved in a complex with FAD in the active site [20]. Fad1p is a single-domain monofunctional enzyme belonging to the 3′-phosphoadenosine 5′-phosphosulfate (PAPS) reductase family and shows little or no sequence similarity to the prokaryotic FAD-forming enzymes. The human gene for FADS, named *FLAD1* [14,21], generates different alternatively spliced transcripts encoding different isoforms, most of them containing the PAPS reductase domain fused with an N-terminal molybdopterin-binding (MPTb) domain with FAD hydrolyzing activity [22].

Our group previously demonstrated that *S. cerevisiae* mitochondria are able to catalyze FAD hydrolysis via enzymatic activity, which is different from the already characterized NUDIX hydrolases and regulated by the mitochondrial NAD redox status [23].

The relevance of *FLAD1*, postulated in our previous papers [24,25], emerged in 2016 when *FLAD1* was identified as a novel Riboflavin-responsive MADD disease gene [26]. MADD defines a heterogeneous class of lipid storage myopathies associated with impaired fatty acid, amino acid and choline metabolisms [27]. In the frame of these studies, a novel isoform containing the sole PAPS domain, hFADS6, was identified and then characterized in some detail [28]. The homology modeling of the PAPS reductase domain of human FADS was performed using the orthologue from *Candida glabrata* as a template [28].

The simultaneous presence in the same cell of the different *FLAD1* transcripts is in line with a possible different subcellular localization for the different isoforms in mammals. Consistently, the existence of distinct cytosolic, nuclear and mitochondrial FADS was demonstrated in different rat models [29,30]. The human transcript variant 1 encodes a 65.3 kDa protein, hFADS1, which contains a predictable mitochondrial-targeting peptide [21]. Mitochondrial localization of hFADS1 was proven in vitro by mitochondrial import assay and confocal microscopy [29].

Mitochondrial supply of FAD is crucial also in yeast, where it is expected to be delivered to a number of nascent client apo-flavoenzymes [31]. This occurs in some cases, such as for the flavoprotein subunit of succinate dehydrogenase (Sdh1p), via an autocatalytic covalent flavinylation mechanism, which could be assisted by ancillary proteins [32,33,34,35,36].

The first eukaryotic gene encoding for a mitochondrial FAD translocator was identified in *S. cerevisiae* and named *FLX1*, whose human orthologue is *SLC25A32.* Interestingly, mutations in the *SLC25A32* gene have been identified as the cause for RR-MADD [37,38], thus confirming the essential role of flavin cofactor supply for mitochondrial proteome.

The existence of a mitochondrial FADS isoform in yeast is still controversial as well as the physiological role of the mitochondrial FAD translocator, Flx1p. Initially, it was reported that FAD is synthesized by Fad1p exclusively in the cytosol [16] and, consequently, imported into mitochondria via Flx1p [39]. However, results from our laboratory showed that, besides in the cytosol, FAD-forming activities can be specifically revealed in mitochondria, entailing FAD precursors uptake in mitochondria and mitochondrial FAD export to cytosol via Flx1p [40,41]. Moreover, FAD movement across the mitochondrial membrane catalyzed by Flx1p plays an additional regulatory role on apo-Sdh1p biogenesis at the post-transcriptional level [42].

As a matter of fact, the protein responsible for FAD synthesis in *S. cerevisiae* mitochondria remains to be identified and characterized. In this paper, we report the existence of *S. cerevisiae* Fad1p as two distinct echoforms localized to both cytosol and mitochondria, and the presence of two populations of *FAD1* mRNAs, which differ for their 3′UTRs. Whether and how the 3′UTRs play a role in the mechanism that destines Fad1p to either cytosol or mitochondria will be dealt with.

## 2. Materials and Methods

### 2.1. Materials

All reagents and enzymes were from Sigma-Aldrich (St. Louis, MO, USA), ThermoFisher Scientific (Waltham, MA, USA) and Merck KGaA (Darmstadt, Germany). Zymolyase was from MP Biomedicals (Aurora, OH, USA). Bacto Yeast Extract and Yeast Nitrogen base were from BD Difco (Franklin Lakes, NJ, USA). Monoclonal antibody against STREP-tag (α-STREP) was obtained from IBA Lifesciences (Göttingen, Germany), monoclonal antibody against actin (α-ACTIN) was from Abcam (Cambridge, MA, USA), secondary anti-rabbit or anti-mouse IgG antibodies conjugated with peroxidase were from ThermoFisher Scientific (Waltham, MA, USA). The RNeasy Midi Kit was from Qiagen (Hilden, Germany). The enhanced AMV Reverse transcriptase kit was from Sigma-Aldrich (St. Louis, MO, USA). The Dynabeads mRNA Purification kit was from Invitrogen (Waltham, MA, USA). The Wizard SV Gel and PCR Clean-up system was from Promega (Madison, WI, USA).

### 2.2. Yeast Strains and Recombinant Multicopy Plasmids

The wild-type *S. cerevisiae* strain CEN.PK113-13D (also named EBY157, WT, or K26; genotypes *MAT*α, *MAL2*-*8^c^*, *SUC2* and *ura3*-*52*) was previously described in [40,42]. The CENPK2-1C, derived from the CEN.PK yeast series, was used as a recipient strain to express a recombinant Fad1p carrying a STREP tag at the N- or C-terminal end. To construct the multicopy plasmids p426HXT7-FAD1-STREP, p426-MET25-FAD1-STREP, and p426-MET25-STREP-FAD1, the *FAD1* gene was cloned by PCR with a selected pair of primers designed to amplify the complete *FAD1* open reading frame (ORF), skipping the contiguous 5′ and 3′ UTR regions. The amplified fragments were cloned into the multicopy plasmid p426-HXT7-STREP or p426-MET25-STREP using the standard gap-repair procedure. In these vectors, the recombinant *FAD1* construct, encoding Fad1p fused to the STREP tag (Trp-Ser-His-Pro-Gln-Phe-Glu-Lys) at the N- or C-terminal end, was placed between the *HXT7* or the *MET25* promoter and *CYC1* terminator. The transformation of the CEN.PK113-13D and CENPK2-1C strains with the recombinant or empty vectors was carried out according to the frozen competent cell procedure [43]; the transformed strains were selected for the presence of URA3 genetic marker, which confers the ability to grow in a minimal synthetic liquid medium (SM, 6.7 g/L yeast nitrogen base supplemented with 25 mM Histidine, 0.44 mM Leucin and 0.19 mM Tryptophan) without uracil. All the transformed strains used in this study are summarized in Table 1.

### 2.3. Media and Growth Conditions

The wild-type K26 cells were grown aerobically at 30 °C with constant shaking in rich liquid medium (YEP, 10 g/L yeast extract, 20 g/L Bacto Peptone). The K26-transformed strains were grown aerobically at 30 °C with constant shaking in SM medium. The CENPK2-1C strain and the novel derived strains (see Table 1) were grown aerobically at 30 °C with constant shaking in SM medium. Ethanol, glycerol, galactose or glucose (2% each) were used as carbon sources. The solid medium (YEP or SM) contained 18 g/L agar.

### 2.4. Preparation of Cellular Extracts

Cells grown up to the early exponential phase (5 h) were harvested by centrifugation (8000× *g* for 5 min), washed with sterile water, resuspended in 250 µL of lysis buffer (10 mM Tris-HCl, pH 7.6, 1 mM EDTA, 1 mM dithiothreitol, 0.2 mM phenylmethanesulfonylfluoride (PMSF), supplemented with one tablet of Roche protease inhibitor cocktail every 10 mL of lysis buffer), and vortexed with glass beads for 10 min at 4 °C. The supernatant was removed and centrifuged at 3000× *g* for 5 min to remove cell debris.

### 2.5. Preparation of Spheroplasts, Cytoplasm, Mitochondria, and Submitochondrial Fractions

Spheroplasts were prepared using Zymolyase. Mitochondria and cytoplasm were isolated from spheroplasts as described in [23,40]. Mitochondria, ruptured by osmotic shock, were centrifuged at 20,000× *g* for 30 min to separate the mitochondrial soluble fraction and the mitochondrial membrane-enriched fraction.

### 2.6. Western Blotting

Protein from subcellular fractions or from cellular extracts were separated by SDS-PAGE and transferred onto a PVDF membrane as in [42]. The immobilized proteins were incubated with either a monoclonal antibody against the STREP-tag raised in mouse (α-STREP) to detect the recombinant Fad1p or a polyclonal α-FADS antiserum (2000-fold dilution) raised in rabbit to detect Fad1p [29]. Immuno-reactive materials were visualized with the aid of a secondary anti-rabbit or anti-mouse IgG antibody conjugated with peroxidase. PVDF membranes were also probed with monoclonal α-ACTIN antibody (10,000-fold dilution, raised in mouse), to reveal yeast actin (Act1p).

### 2.7. RNA Isolation

The RNeasy Midi Kit was used to extract total RNA from WT cells grown up to early exponential phase. Total RNA concentration was quantified by monitoring absorbance to 260 nm (OD_260 nm_), RNA integrity was verified by formaldehyde agarose gel electrophoresis and its purity was evaluated by measuring the OD_260 nm_/OD_280 nm_ ratio.

### 2.8. 3′RACE Analysis

3′RACE analysis was performed on poly(A)^+^ RNA obtained from 75 µg of total RNA using the Dynabeads mRNA Purification kit, according to the manufacturer’s instructions. The purified poly(A)^+^ RNA was reverse-transcribed using the Enhanced AMV Reverse Transcriptase Kit and the oligo(dT)-anchor primer [oligo(dT)-AP], according to the manufacturer’s instructions. The cDNA pool was diluted 10-fold and these were used as a template (2.5 µL) for PCR reactions using a gene-specific primer and the AP primer. The specificity of the first round PCR-amplified fragment was proved by performing a semi-nested PCR. The sequence of primers, used in this study, are reported in Table 2. The 3′RACE products were separated by 2% agarose gels. The fragments obtained with the first round PCR were purified from preparative agarose gel by using the Wizard SV Gel and PCR Clean-up system and then sequenced.

### 2.9. Semiquantitative RT-PCR Assay

Semiquantitative RT-PCR assay was performed on 200 ng of total RNA that was reverse-transcribed using the Enhanced AMV Reverse Transcriptase Kit. The cDNA pool was diluted 10-fold and 1 µL was used as a template in PCR-amplification using gene-specific primers (Table 2). Amplification conditions were as follows: 94 °C for 2 min, then 25 cycles of 94 °C for 15 s, 60 °C for 30 s, and 68 °C for 1 min. For each sample, *ACT1* mRNA was used as an internal control. *SDH1* mRNA was used as a control of inducible expression under nonfermentable carbon source growth conditions.

### 2.10. Other Assays

Fumarase (FUM), succinate dehydrogenase (SDH) and phosphoglucoisomerase (PGI) activities were spectrophotometrically measured as in [40,42,44]. Protein concentration was assayed according to Bradford [45], using bovine serum albumin as the standard.

## 3. Results

### 3.1. Mitochondrial Localization of Recombinant Fad1p

Our previous results based on functional approaches showed that, besides in the cytosol, FAD synthesizing activity can be specifically revealed in mitochondria from *S. cerevisiae* [40,46]. This result was somewhat surprising since a single gene, named *FAD1*, was reported as coding for a sole cytosolic FADS [16]. *FAD1* gene, as normally occurs in yeasts, has no introns; thus, different transcripts generated by alternative splicing are not expected. Consistently, a single transcript is reported in the Ensemble database (http://www.ensembl.org/index.html, accessed on 13 September 2021) from which a single translation product, Fad1p, is predictable with NCBI tool ORF finder (www.ncbi.nlm.nih.gov/orffinder, accessed on 13 September 2021). No mitochondrial-targeting peptide could be found in this product using several prediction programs, such as iPSORT, MitoProt, TargetP and PSORT II. Concerning these aspects, the yeast *FAD1* differs from the human *FLAD1* gene, which generates alternatively spliced isoforms with different subcellular localizations [29,30]. Nevertheless, one of these isoforms, hFADS6, strongly resembles the yeast protein Fad1p, showing a high level of conservation for the PAPS reductase domain (Figure 1).

To gain some insight into the cellular destination of Fad1p, we investigated the subcellular localization of the recombinant form of Fad1p carrying the STREP-tag at the C-terminus and overexpressed in the CENPK2-1C strain (Table 1). Therefore, subcellular fractions (spheroplasts, mitochondria and cytoplasm) were prepared from CENPK2-1C^FAD1-STREP^ cells grown in ethanol up to early exponential growth phase and equal amounts of each fraction were analyzed by immunoblotting using the α-STREP antibody (Figure 2a). Even though the recombinant Fad1p–STREP was enriched in the cytoplasmic fraction, an evident immuno-reactive band was observed in the mitochondrial fraction, suggesting a mitochondrial localization for the recombinant Fad1p–STREP. Interestingly, the recombinant Fad1p–STREP was immunodecorated at the same size both in mitochondria and in cytoplasm. When the STREP-tag was moved from the C- to the N-terminal end, the ability of the recombinant STREP–Fad1p to enter mitochondria was completely lost, accompanied by a significant enrichment in the cytoplasmic fraction (Appendix A).

To further prove the ability of the recombinant Fad1p to enter mitochondria, we placed Fad1p–STREP expression under the control of HXT7 promoter in the multicopy plasmid p426-HXT7-FAD1-STREP (Table 1) and used a different genetic strain, namely, K26. Subcellular fractions were prepared from K26^FAD1-STREP^ cells grown in ethanol at the early exponential phase and analyzed by immunoblotting carried out with the α-STREP (Figure 2b). The recombinant protein Fad1p–STREP was immunodecorated at the same size in both cytoplasm and mitochondria, where it was significantly enriched. When the K26^FAD1-STREP^ cells were in the presence of the uncoupler carbonyl cyanide-p-trifluoromethoxy-phenylhydrazone (FCCP), the Fad1p–STREP import into mitochondria was prevented (Figure 2b), as expected for a process depending on mitochondrial membrane potential [47]. This finding clearly indicates that the recombinant Fad1p is imported and localized to mitochondria.

### 3.2. Natural Fad1p Exists in Mitochondria

To ascertain whether the activity of FADS that we detected inside mitochondria is actually accountable to Fad1p, we analyzed the subcellular localization of natural Fad1p in cells grown either on nonfermentable (glycerol and ethanol) or fermentable (glucose and galactose) carbon sources up to the early exponential phase (5 h) (Figure 3a). A Fad1p band was detected in mitochondria prepared from glycerol-grown cells by using the α-FADS antiserum. It should be noted that even though actin (Act1p) was used as a cytosolic marker [48], a minimal amount of Act1p was also found to be associated to the mitochondrial fraction in agreement with [49]. In ethanol, an even more intense band was detected in the mitochondrial fraction, whereas in glucose-grown cells, the mitochondrial Fad1p level was very low. Thus, the Fad1p content in mitochondria seems to depend on the carbon source. As a control, the dependence of the mitochondrial SDH specific activity on different carbon sources was analyzed (Figure 3b), thus verifying the expected glucose repression due to both transcriptional and post-transcriptional events [50,51]. It is noteworthy that in the mitochondria prepared from galactose-grown cells, no band was detected (Figure 3a), even though we observed a high (induced) level of SDH activity (Figure 3b), in agreement with the presence of mitochondria in cells cultured in galactose as the carbon source [52]. Differently from what was observed for the mitochondrial Fad1p, the cytosolic echoform did not appear to depend on the carbon source (Figure 3a).

Cellular sublocalization of Fad1p was also evaluated in K26 cells grown in glycerol up to the stationary phase (24 h), as in [40]. Cytoplasmic and mitochondrial fractions were prepared from spheroplasts and equal amounts of each fraction were used to reveal Fad1p by immunoblotting analysis (Appendix A). A faint α-FADS immuno-reactive band was found in the cytoplasmic fraction. A much more evident band migrating at the same Mr was observed in total mitochondrial fraction, consistently enriched with respect to the spheroplast specific amount. A similar enrichment was observed for the fumarase (FUM) and succinate dehydrogenase (SDH), two mitochondrial markers (Appendix A, histogram). The absence of cytosolic contamination in mitochondria was demonstrated by the absence of phosphoglucoisomerase (PGI) activity, a marker of cytosolic fraction (Appendix A, histogram). Isolated mitochondria were further subfractionated in a membrane-enriched fraction (M_fr_) and in a soluble fraction (S_fr_). The Fad1p band was clearly detectable in both submitochondrial fractions, whose purity was confirmed by measuring the inner membrane marker SDH and the matrix marker FUM enzymatic activities (Appendix A, histogram).

Overall, on a molecular basis, these results support our previous functional data, proving that Fad1p exists in yeast mitochondria. Since no difference in size was observed between cytoplasmic and mitochondrial Fad1p, we postulated that the *FAD1* gene might generate two identical echoforms, destined to two distinct subcellular localizations.

### 3.3. Detection of Two FAD1 mRNAs with Different 3′UTR and Their Carbon Source Dependence Profile

In order to unravel the mechanism responsible for the subcellular destination of Fad1p, we considered a series of indications derived from the literature, which proposes a role for *cis*-acting 3′UTR elements in targeting transcripts to mitochondria [53,54], and from a genome-wide analysis, which suggests a high probability for *FAD1* mRNA to be located on mitochondria-bound polysomes [54]; hence, the protein would be imported while being translated in a co-translational process [55,56,57]. Therefore, we hypothesized that important information for Fad1p localization could reside in the 3′UTR of *FAD1* mRNA. Since no information was available, a combination of bioinformatic analyses and experimental procedures were carried out to gain knowledge about this untranslated region.

Firstly, we searched for putative polyadenylation sites (PASs), despite their complexity [58,59], and found two canonical signals (ATTAAA) in the 1 kbp region downstream of *FAD1* ORF using DNAFSMiner [60]. The proximal one was placed 50 nucleotides after the stop codon and the distal one was placed 339 nucleotides after the stop codon. The bioinformatic tool also predicted other different PASs with similar scores, namely, AATAAA and TATCAA (Appendix A). These predictions suggested that *FAD1* mRNA may be present in different forms, whose length depends on the alternative use of distinct PASs in the transcript 3’UTR.

To test this hypothesis, 3′RACE experiments were performed on a template cDNA obtained from purified poly(A)^+^ RNA prepared from WT cells grown at the early exponential phase on glucose as a carbon source and the gene-specific primers (Table 2; Figure 4a). When the primers FAD1A.for and AP were used, a single amplified product of about 400 bp was obtained (Figure 4b, lane 1). The specificity of the 400 bp product was verified by a semi-nested PCR using the gene-specific primer FAD1C.for located 91 nucleotides downstream of FAD1A.for. The amplified product was obtained at the expected size, which is about 300 bp (Figure 4b, lane 2). The 400 bp fragment was sequenced, revealing the existence of *FAD1* transcript with a 3′UTR of 128 nucleotides (Appendix A). To search for the existence of a longer 3′UTR region of *FAD1* mRNA, a PCR was performed using the primers FAD1C.for and FAD1B.rev, the last being located downstream of the end of the 3′UTR previously identified (Figure 4b); in this case, a product of about 450 nucleotides was obtained (Figure 4b, lane 3), indicating the existence of a longer *FAD1* mRNA. To define the entire length of the 3′UTR of this transcript, a PCR reaction was performed using the primers FAD1B.for and AP, obtaining a product of 550 bp (Figure 4b, lanes 4 and 5). Its specificity was proved by a semi-nested PCR with the primers FAD1B.for and FAD1B.rev, which gave rise to a fragment of about 100 bp, as expected (Figure 4b, lane 6). The 550 bp product was sequenced, showing the existence of an additional *FAD1* mRNA with a longer 3′UTR region of 759 bp (Appendix A). Similar results were obtained when the analysis was performed using cDNA as a template, obtained starting from purified poly(A)^+^ RNA prepared from WT cells grown on glycerol (Appendix A). Thus, *FAD1* generates (at least) two *FAD1* mRNAs with different 3′UTR lengths. Based on the different length of the 3′UTR, we named them ‘short *FAD1* mRNA’ and ‘long *FAD1* mRNA’, respectively.

To understand whether or not the length of 3′UTR is responsible for the fate of Fad1p echoforms, the carbon source dependence of the *FAD1* transcript levels was investigated. To this purpose, total RNA was extracted from WT cells grown at the early exponential phase on different carbon sources, and the amount of *FAD1* transcripts was evaluated by semiquantitative RT-PCR, with *ACT1* mRNA used as an internal control, essentially as in [42]. The long *FAD1* mRNA was amplified using the primer pair FAD1A.for and FAD1B.rev, whereas the total *FAD1* mRNA (short + long) was amplified using the primer pair FAD1A.for and FAD1A.rev (Appendix A). The RT-PCR product (540 bp) relative to the long *FAD1* mRNA complemented the 3′RACE data, confirming the presence of a longer transcript. The relative amount of total *FAD1* mRNA did not depend on the carbon source, whereas a carbon source dependence was observed for the long *FAD1* mRNA: its amount was higher in ethanol grown cells, reduced in cells grown on glucose and almost absent in galactose-grown cells. As a control, the expression pattern of *SDH1* mRNA was analyzed; it was repressed by glucose, but not by galactose (Appendix A), in agreement with SDH activity reported here (Figure 3, histogram) and in literature [50].

### 3.4. A Mitochondrial Localization Motif in FAD1 3′UTRs

To further confirm that Fad1p localization mechanism might be influenced at the mRNA level, we searched for *cis*-acting elements that could be responsible for targeting *FAD1* mRNA to the outer mitochondrial membrane. Sequence inspection revealed the existence of a putative *cis*-acting motif TGTATATACA containing the consensus mitochondrial localization motif M1 (TGTA(a/c/t/)ATA), as defined in [61], and resembling the motif 6 (WTATWTACADG) reported as a mitochondrial downstream motif [62]. This element was located about 90 nucleotides downstream the *FAD1* ORF stop codon; therefore, it was present in both *FAD1* transcripts. The same motif (or similar sequences) was not found in either the upstream *FAD1* ORF region or the *FAD1* ORF.

The 1 kbp genomic downstream region *FAD1* ORF contains, on the opposite strand the *MRP10* gene, encoding a mitochondrial ribosomal protein of 10.7 kDa [63,64]. Mrp10p lacks a mitochondrial-targeting sequence (as revealed by bioinformatic analysis); however, the identified M1 mitochondrial localization motif is present in its 3′UTR, as revealed by the Shalgi’s database [61]. This motif is placed 56 nucleotides downstream of the *MRP10* ORF stop codon. It should be noted that the M1 motif in 3′UTR of *FAD1* and *MRP10* is a palindromic sequence that could be read on both DNA strands (Figure 5). This property implies a rather nontrivial extension of the M1 motif and allows the same region to play the role of mitochondrial localization motif for both the *MRP10* and the *FAD1* transcripts.

## 4. Discussion

Experiments reported in this paper point out the issue of the origin of mitochondrial FAD in *S. cerevisiae* mitochondria, strictly related with the problem of the identity and of the subcellular localization of the FAD forming enzyme, FADS. Immunoblotting experiments reported here reveal the presence of an α-FADS immuno-reactive band in mitochondria, showing the same Mr of that found in the spheroplasts and cytoplasm. This is in line with the absence of a putative cleavable presequence in Fad1p, the sole peptide generated by *FAD1* gene. Thus, these results prove, on a molecular basis, Fad1p localization to yeast mitochondria and support the absence of a cleavable mitochondrial signal peptide in Fad1p.

It is well-documented that in eukaryotic cells, a protein can be located to two different subcellular compartments; this dual localization results in different echoforms—that is, proteins with identical or nearly identical amino acid sequences distinctly placed in the cell [65,66,67]. This is an important and frequent phenomenon; in fact, more than one third of the yeast mitochondrial proteome seems to be dual localized [68,69]. It has also become evident that dual localization can be regulated, induced and rebalanced in response to either cellular signaling or changing extracellular conditions. Different dual-targeting mechanisms of mitochondrial proteins have been described or suggested, and, in some case, they seem to be tightly regulated in time, location and function [65]. Our results demonstrate that *FAD1* generates two echoforms.

Analyzing the dependence of the Fad1p levels on fermentable and nonfermentable carbon sources (Figure 3), we verified that the amount of Fad1p in mitochondria, but not in the cytoplasm, depends on the carbon source being higher in ethanol and glycerol than in the fermentable glucose. Interestingly, mitochondrial Fad1p almost disappeared in galactose-grown cells. This finding may settle the discrepancy between results from our (see Introduction) and other laboratories [32,39] about the existence of mitochondrial FADS and, consequently, about the direction of FAD transport via the inner mitochondrial membrane translocator Flx1p. The absence of mitochondrial Fad1p, observed in galactose-grown cells, implies that in this condition FAD must be transported from cytosol to mitochondria via Flx1p (Figure 3 [32,39]). Vice versa, in a nonfermentable carbon source and in glucose, in the presence of mitochondrial Fad1p, Flx1p is expected to mediate FAD efflux from mitochondria to cytosol acting as a regulator of apo-flavoprotein biogenesis [40,42].

Using yeast strains overexpressing a recombinant form of Fad1p, the dual location of natural Fad1p was confirmed. Our data demonstrate that Fad1p is, per se, able to enter mitochondria, despite the absence of a canonical mitochondrial-targeting sequence. A “mitochondrial destination” message should be localized at the N-terminal end, since Fad1p loses its ability when the STREP-tag is moved from the C- to the N-terminal end (Appendix A). Presumably, the α1-helix in the first twenty amino acids at the N-terminal end, as well as the contiguous α2-helix, indeed have amphipathic characteristics, with the basic residues exposed to the solvent from the same side of the central axis, as revealed by the crystal structure solved at 1.9 Å resolution (PDB entry: 2wsi) [20]. This would be in agreement with the involvement of this protein moiety in the import process. Alternatively, another putative noncanonical/weak mitochondrial-targeting sequence [70] could be implicated in the mitochondrial destination of Fad1p.

We also observed that in galactose-grown cells, the recombinant Fad1p-STREP entered mitochondria per se [71], behaving differently from the natural Fad1p that was unable to enter mitochondria. All together, these data suggest the existence of a fine mechanism regulating the localization of the natural Fad1p, presumably at the transcript level and dependent on the carbon source. It is well-established that intracellular localization of mRNA has a significant impact on the efficiency of its translation and, presumably, determines its response to changing metabolic conditions or cellular stress [72].

Our attention was focused on the 3′UTR region of *FAD1* mRNA. Since no information was available on this region, we carried out a 3′RACE analysis and found the existence of (at least) two *FAD1* mRNAs with different 3′UTR lengths (Figure 4). This finding could be relevant for Fad1p cell biology, since it is well-known that 3′UTR regulates multiple aspects of mRNA metabolism, including subcellular localization, translation efficiency and stability, in cooperation with different RNA binding proteins [73]. A mechanism that could have a role in generating different transcripts with specific subcellular localizations and/or functions might be alternative polyadenylation [74,75]. We predicted different canonical and noncanonical putative PASs localized in the first 1 kbp downstream of *FAD1* ORF. Whether or not these PASs are actually active and responsible for the generation of the two *FAD1* transcripts are matter of ongoing investigation in our laboratory.

Searching for possible *cis*-acting motifs responsible for the peripheral mitochondrial localization of *FAD1* transcripts, we revealed the existence of the mitochondrial localization motif M1, placed 87 nucleotides downstream the *FAD1* ORF stop codon (Figure 5). The same motif has been identified within a set of genes encoding mitochondrial proteins, whose mRNAs are translated by polyribosomes attached to the cytosolic side of the outer mitochondrial membrane [76]. This motif is present in both *FAD1* transcripts, which might localize close to the mitochondrial outer membrane. Intriguingly, this finding is in agreement with a value of mitochondrial localization of mRNA (MLR) for *FAD1* equal to 87 [54]. This value is similar to the MLR values measured for other mRNAs, whose products have been demonstrated to be internalized into mitochondria (i.e., 87.8 for *SDH1* transcript), thus indicating that Fad1p translation has a high probability to occur on the mitochondrial outer membrane. Recently, it has been reported that mRNAs that initiate translation away from mitochondria experience a significant drop in mobility toward the outer mitochondrial membrane and tend to remain there [77]. The M1 motif is a candidate binding site of the RNA binding protein Puf3p [78], localized on the cytosolic side of the outer mitochondrial membrane, where it interacts with some components of the mitochondrial protein import machinery [57,79]. Besides its role in mRNA targeting to mitochondria, Puf3p controls mRNA stability and translation efficiency according to growth conditions and in response to oxidative stress [80,81,82]. The identified palindromic M1 motif is shared by the 3′UTR of *FAD1* transcripts and the 3′ downstream region of *MRP10* gene, which is transcribed in the opposite direction with respect to *FAD1* gene. Since an interaction between the M1 of *MRP10* mRNA and Puf3p has been proven [83,84], an involvement of Puf3p in mitochondrial localization of Fad1p could be expected. By analyzing the carbon source dependence profile of *FAD1* transcripts (Appendix A), we found that the long transcript is expressed in ethanol and in glycerol at a higher level than in glucose; it is almost absent in galactose. The comparison between this profile and mitochondrial Fad1p levels (Figure 3) shows that the carbon source dependence of the long *FAD1* mRNA level strictly parallels the carbon source dependence of the mitochondrial Fad1p level. This observation suggests that the long *FAD1* mRNA might be responsible for the generation of Fad1p mitochondrial echoform. Therefore, although the M1 motif is present in both the short and long *FAD1* transcripts, we propose that the different length of the 3′UTRs could be responsible for the distinct subcellular destinations of Fad1p. This diverse destination could be explained by a differential Puf3p–M1 interaction caused by the exposure of M1 motif in different RNA contexts. In this scenario, the secondary structure of the two 3′UTRs may be causative of the selective interaction with Puf3p. Moreover, the recruitment of additional RNA binding proteins, presumably involved in mRNA decay or Puf3p interaction with mitochondrial import machinery [85], could also be affected by 3′UTR length and structure. Therefore, Puf3p might regulate the dynamic balance between translation efficiency, protein folding and import rate, thus establishing whether Fad1p may be released in the cytosol or imported into mitochondria [81]. Additionally, it has been shown that cytosolic ribosomes have the ability to ‘sense’ features relative to nascent polypeptide chains; between those features, the sequence and structure of 3′UTR mRNAs can influence the polypeptide synthesis rate, modulating translation accordingly [76,86]. It is noteworthy that the M1 motif was not included in the constructs used for the expression of recombinant Fad1p. Nonetheless, the protein was capable, per se, of localizing to mitochondria in the presence of the mitochondrial membrane potential (Figure 2; [71]). Therefore, the M1 sequence would serve to fine-tune the cellular localization of Fad1p echoforms in various metabolic conditions, such as different carbon sources.

Our hypothesis is summarized in Figure 6. We speculate that, when Puf3p is bound to the M1 motif present in the long 3′UTR context (panel a), Fad1p synthesis efficiency and/or folding rate are slowed down, allowing the co-translational import to proceed and generate the Fad1p mitochondrial echoform. Vice versa, when Puf3p interacts with the M1 in the short 3′UTR (panel b), a higher translation rate might cause a premature folding, thus favoring the cytosolic destination of Fad1p.

## Figures and Tables

**Figure 1 life-11-00967-f001:**
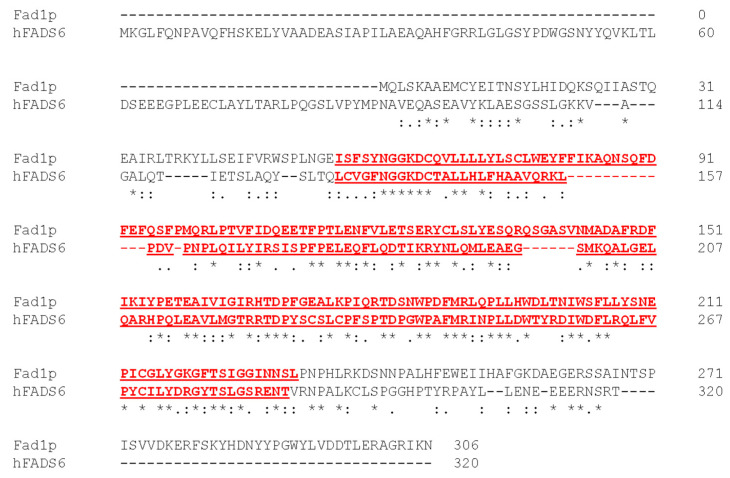
Alignment of FAD synthase sequences from yeast and humans. The alignment of yeast Fad1p with human FAD synthase isoform 6 (hFADS6) was performed by the *Pairwise Sequence Alignment* tool available at the EMBL-EBI website. The PAPS reductase domain (InterPro accession: IPR002500), present in both *S. cerevisiae* Fad1p (amino acids 55-231) and hFADS6 (amino acids 132-286), is underlined and colored in red. Asterisks depict identical residues in the same positions, whereas colons and periods indicate strongly and weakly similar residues, respectively.

**Figure 2 life-11-00967-f002:**
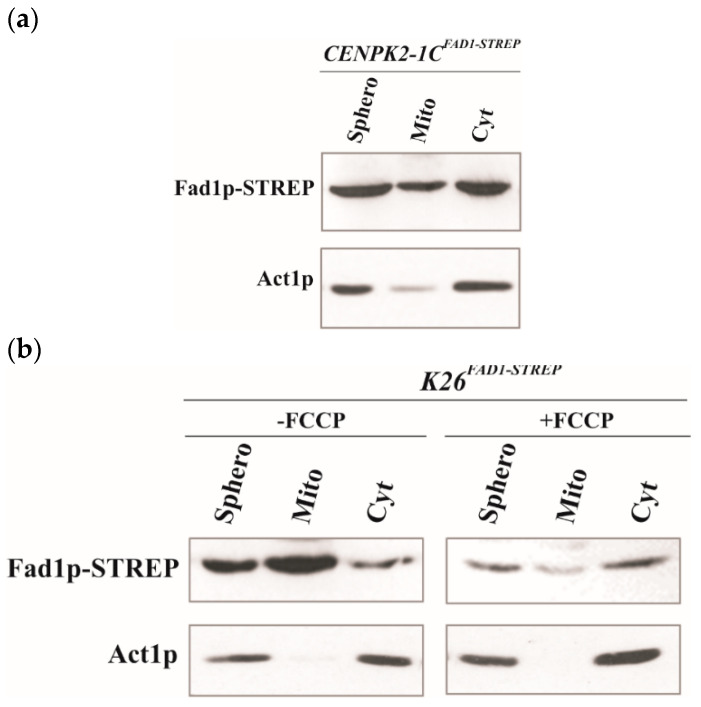
Mitochondrial localization of recombinant Fad1p. Spheroplasts (Sphero), mitochondria (Mito) and cytoplasm (Cyt) were prepared from CENPK2-1C (**a**) and K26 (**b**) cells transformed with the multicopy recombinant plasmid (Table 1). Both strains were grown up to the early exponential phase on ethanol. K26^FAD1-STREP^ cells were incubated in the absence or presence of FCCP (25 µM). Proteins (0.1 mg) were separated by SDS-PAGE, transferred onto PVDF membrane and detected with either α-STREP or α-ACTIN antibodies.

**Figure 3 life-11-00967-f003:**
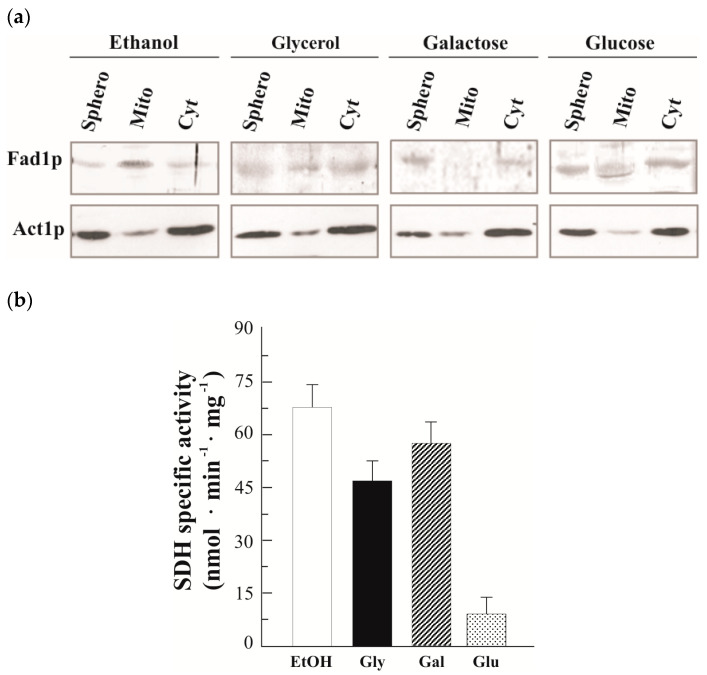
Subcellular distribution of Fad1p in different carbon sources. (**a**) Spheroplasts (Sphero), mitochondria (Mito) and cytoplasm (Cyt) were prepared from WT cells grown up to the early exponential phase at 30 °C in YEP liquid medium supplemented with the indicated carbon sources (2% each). Proteins (0.1 mg) were separated by SDS-PAGE, transferred onto PVDF membrane and detected with either α-FADS or α-ACTIN antibodies. (**b**) The carbon source dependence of the mitochondrial SDH-specific activity is reported as the means (±SD) of two independent measurements.

**Figure 4 life-11-00967-f004:**
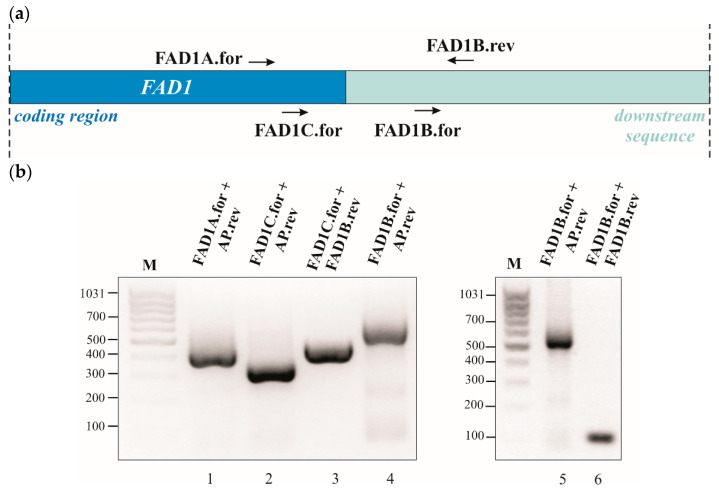
Characterization of *FAD1* mRNA 3′UTR by 3′RACE analysis. (**a**) *FAD1* coding region and 1 kbp downstream region are schematized. Arrows designate the position and direction of gene-specific primers used for 3′RACE. (**b**) The experiment was carried out on a poly(A)^+^ RNA prepared from WT cells grown up to the early exponential phase at 30 °C on glucose. PCR products were separated by a 2% agarose gel. Fragments in lanes 1 and 4 were gel-extracted, purified and sequenced (Appendix A). On the left side of each gel, band sizes (bp) of the marker [M, *MassRuler Low Range DNA Ladder*, ThermoFisher Scientific (Waltham, MA, USA)] are indicated.

**Figure 5 life-11-00967-f005:**
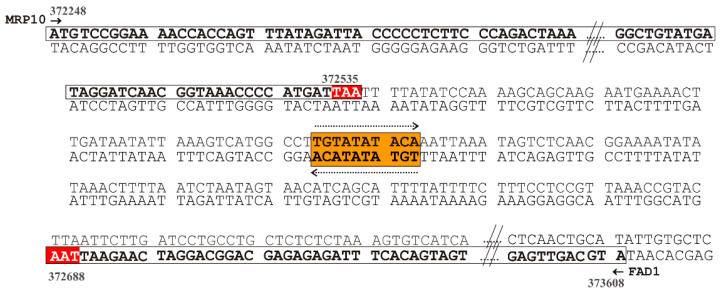
The *cis*-acting motif M1 in the 3′UTR of *FAD1* transcripts. Schematic representation of the partial sequence of the *S. cerevisiae* genomic region comprised between nucleotides 372248 and 373617 (chromosome IV). The white boxes indicate *FAD1* ORF on the Crick strand and *MRP10* ORF on the *Watson* strand; the orange box indicates the mitochondrial localization motif M1, placed about 87 and 55 nucleotides downstream of the stop codon (TAA, highlighted in red) of *FAD1* and *MRP10* ORFs, respectively. The dotted arrows highlight the bidirectionality of the M1 palindromic sequence.

**Figure 6 life-11-00967-f006:**
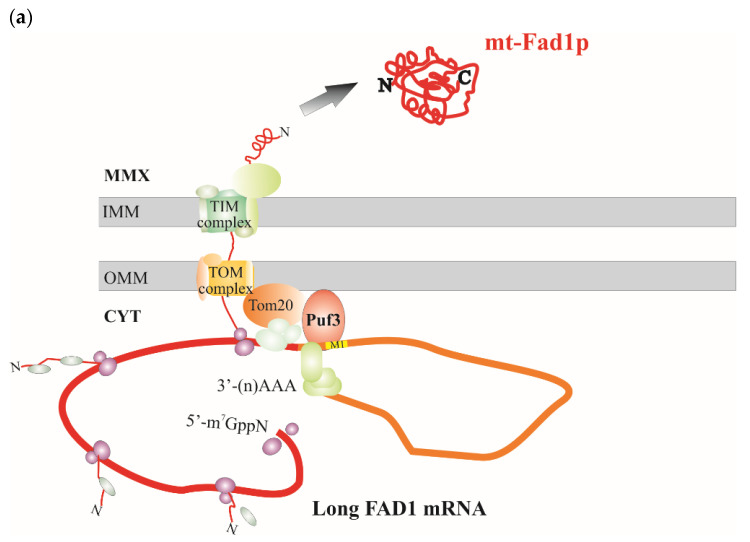
Model depicting the dual location of Fad1p. The cartoon shows a possible role of Puf3p in the different destination of Fad1p echoforms. (**a**) Puf3p binds to the long *FAD1* mRNA and drives the nascent polypeptide in the mitochondrial matrix (MMX) via TOM–TIM complexes before the fully translated protein starts to fold. (**b**) Puf3p binds to the short *FAD1* mRNA. As translation is presumably not sufficiently slowed down, premature and rapid folding of the nascent polypeptide in the cytosol (CYT) prevents Fad1p import across mitochondrial membranes (OMM—outer mitochondrial membrane; IMM—inner mitochondrial membrane).

**Table 1 life-11-00967-t001:** *S. cerevisiae* strains used in this study.

Strain	Multicopy Plasmid
CEN.PK113-13D (EBY157, WT, K26)	/
K26 ^CTR^	p426-HXT7-STREP
K26 ^FAD1-STREP^	p426-HXT7-FAD1-STREP
CENPK2-1C ^FAD1-STREP^	p426-MET25-FAD1-STREP
CENPK2-1C ^STREP-FAD1^	p426-MET25-STREP-FAD1

**Table 2 life-11-00967-t002:** Primers used in this study.

Gene	Primer	Sequence
*FAD1*	FAD1A.for	5′-ATCGGCGGAATTAACAACTCA-3′
	FAD1A.rev	5′-TTGCCAAATGCATGAATGATTT-3′
	FAD1B.for	5′-GCCTAGCGGC GTGATAGTTAA-3′
	FAD1B.rev	5′-TGCTGGCTTAGTAACGGAATTG-3′
	FAD1C.for	5′-CATTTGGCAAGGACGCAGAA-3′
*SDH1*	SDH1.for	5′-GCCAATTCCTTGTTGGATCTTG-3′
	SDH1.rev	5′-TGGCAACCCAGGCTGTAAAG-3′
*ACT1*	ACT1.for	5′-TTCCATCCAAGCCGTTTTGT-3′
	ACT1.rev	5′-GGCGTGAGGTAGAGAGAAACCA-3′
	Oligo(dT)-AP	5′-GACCACGCGTATCGATGTCGAC(T)_16_V-3′
	AP.rev	5′-GACCACGCGTATCGATGTCGAC-3′

## Data Availability

Not applicable.

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
