# Peer review of "Subcellular Localization of Fad1p in Saccharomyces cerevisiae: A Choice at Post-Transcriptional Level?"

_life, 2021, doi:10.3390/life11090967_

Round 1

Reviewer 1 Report

Very good work, ir brings new interesting data to the field and may have impacto on human studies 

Author Response

Very good work, ir brings new interesting data to the field and may have impacto on human studies 

We would like to thank the Reviewer for emphasizing the value of the manuscript in the mitochondrial field.

Reviewer 2 Report

This was a well-done study providing strong evidence that the mRNA for Fad1 in yeast has two echoforms which help localize the Fad1 protein to either the cytosol or mitochondria. The experiments were appropriate and manuscript was well written. My only concern was that their hypothesis (culminating in the model shown in Figure 6) is left unresolved. Why couldn't they remove the "mitochondrial destination sequence" in the 3' UTR of the long form and determine whether or not the protein localizes to mitochondrial This does not seem to be such a difficult experiment, so I don't understand why it wasn't done. I strongly suggest that the authors perform this one last experiment. Apart from this, I'm supportive of publishing the study. 

Author Response

1) This was a well-done study providing strong evidence that the mRNA for Fad1 in yeast has two echoforms which help localize the Fad1 protein to either the cytosol or mitochondria. The experiments were appropriate and manuscript was well written.

We thank the Reviewer for expressing this gratifying opinion.

2) My only concern was that their hypothesis (culminating in the model shown in Figure 6) is left unresolved. Why couldn't they remove the "mitochondrial destination sequence" in the 3' UTR of the long form and determine whether or not the protein localizes to mitochondrial This does not seem to be such a difficult experiment, so I don't understand why it wasn't done. I strongly suggest that the authors perform this one last experiment. Apart from this, I'm supportive of publishing the study. 

The Reviewer makes a good point and we apologise for not having adequately clarified this aspect. The experiment requested by the Referee was carried out with the recombinant protein (Figure 2). As described in the Materials and Methods section, the plasmids p426HXT7 and p426MET25 (used for the expression of recombinant Fad1p) contained the complete ORF but not the UTRs; therefore, the mitochondrial destination motif M1 was not present in these versions. The data reported in Figure 2 clearly show that the protein is per se capable of localizing to mitochondria in the presence of the mitochondrial membrane potential. The M1 motif provides to address FAD1 transcripts to the outer mitochondrial membrane with the help of specific RNA binding proteins, such as Puf3, so facilitating the mitochondrial import of Fad1p translated from the long FAD1 mRNA. Ultimately, the M1 sequence would serve to fine-tune the cellular localization of Fad1p echoforms in the presence of various metabolic conditions, such as different carbon sources. To make this point clearer, we have now amended the Methods (paragraph 2.2, lines 142-143) and the Discussion (lines 507-512).

Reviewer 3 Report

This manuscript by Bruni and colleagues looks at flavin adenine dinucleotide (FAD) synthase in yeast (FAD1 gene expressed as Fad1p), which catalyzes the last step in synthesis of FAD from riboflavin. The introduction gives reasonable background on the synthesis of FAD and the group’s previous work on demonstrating that FAD can be synthesized both in the cytosol and mitochondria of yeast cells. The authors make connections where appropriate to mammalian FAD synthesis, especially in the formation of succinate dehydrogenase (SDH) as well as related mutations resulting in human disease. Previous data suggested the localization of Fad1p to mitochondria based on results showing that FAD can be synthesized in mitochondria after uptake of precursors and then exported to cytosol via Flx1 (a mitochondrial FAD transporter). Fad1p has not previously been shown to localize to the mitochondria, and it has no canonical mitochondrial targeting sequence (MTS).

One of the primary findings of the manuscript is identifying two echoforms of Fad1p, one in the cytosol and one in the mitochondria.  Figure 2 shows the C-term STREP tagged Fad1p observed in both cytosolic and mitochondrial fraction at the same size band. In separate cell line, Fad1p-STREP mitochondrial localization is shown to be dependent on an intact mitochondrial membrane potential.  Interestingly, Figure S1 shows the N-term STREP tagged Fad1p is not observed in the mitochondrial fraction but is observed in cytosolic fraction.

Although these results are reasonable, there is no mitochondrial control shown in the western blots. The authors should consider using something like anti-VDAC/Porin on the western membranes to show an appropriate mitochondrial control for the following figures:

  • Figure 2, both panels (a) and (b)
  • Figure 3, panel (a)
  • Figure S1
  • Figure S2 (this figure also does not include the cytosolic control anti-ACTIN; both controls should be included in addition to the specific activity data presented)

Figure 3 shows endogenous Fad1p mitochondrial localization is dependent on carbon source where more Fad1p is mitochondrial localized when grown in non-fermentable carbon sources, and a down regulation is seen when grown on fermentable carbon sources. This is further supported by SDH activity in panel (b). Galactose is an interesting result in that NO Fad1p is observed at all in mitochondria, yet SDH activity is high. The authors do a reasonable job of describing and explaining this result.

The other primary finding of this manuscript is the echoform localization of Fad1p is likely achieved by at least two populations of FAD1 mRNAs with differences in the 3’ UTR length. The authors identify a short 3’ UTR and a long 3’ UTR (Figure 4) and show that total FAD1 mRNA did not depend on carbon source but the amount of long FAD1 mRNA depended on the carbon source (Figure S5).

Based on these data, the authors conclude the 3’ UTR is important for the localization of the mitochondrial echoform. It is not clear though if the FAD1 ORF used in the STREP containing plasmids contains the 3’ UTR, or if the mitochondrial localization observed using these cells is a result of overexpression and the N-terminus of Fad1p. The endogenous detection of Fad1p is much more convincing, but this figure would benefit from additional controls (as mentioned above). In the discussion the authors speculate that Fad1p has a “mitochondrial destination” message in the N-terminus of the protein as it was required for the STREP-Fad1p mitochondrial localization. Further explanation based on the crystal structure of the protein or other relevant information would help clarify this.                                                         

The mechanism regulating localization of natural Fad1p at the transcript level dependent on the carbon source is an ongoing investigation. The authors show that the FAD1 3’ UTR is shared with 3’ UTR for MRP10; the two genes share a M1 mitochondrial localization motif (Figure 5). Mrp10p also lacks a canonical MTS like Fad1p. It was previously demonstrated that MRP10 mRNA interacts with Puf3 via its M1 3’ UTR sequence. Puf3 is an important protein for localizing mRNAs to mitochondrial outer membrane for co-translational insertion into mitochondria. The authors speculate that this mechanism is similar for FAD1 mRNA regulation (Figure 6 – proposed model).

Specific comments regarding the manuscript:

Line 48 – Yeasts, as well as fungi, …

Consider changing to “Yeast, as well as other fungi, …” as yeast are a type of fungi and this would clarify the statement.

Line 372 – Figure S2 should be changed to Figure S3

Author Response

This manuscript by Bruni and colleagues looks at flavin adenine dinucleotide (FAD) synthase in yeast (FAD1 gene expressed as Fad1p), which catalyzes the last step in synthesis of FAD from riboflavin. The introduction gives reasonable background on the synthesis of FAD and the group’s previous work on demonstrating that FAD can be synthesized both in the cytosol and mitochondria of yeast cells. The authors make connections where appropriate to mammalian FAD synthesis, especially in the formation of succinate dehydrogenase (SDH) as well as related mutations resulting in human disease. Previous data suggested the localization of Fad1p to mitochondria based on results showing that FAD can be synthesized in mitochondria after uptake of precursors and then exported to cytosol via Flx1 (a mitochondrial FAD transporter). Fad1p has not previously been shown to localize to the mitochondria, and it has no canonical mitochondrial targeting sequence (MTS).

The authors genuinely appreciate the careful analysis of the manuscript performed by the Reviewer and would like to thank her/him for the constructive comments. Below is a list of point-by-point responses.

1) One of the primary findings of the manuscript is identifying two echoforms of Fad1p, one in the cytosol and one in the mitochondria.  Figure 2 shows the C-term STREP tagged Fad1p observed in both cytosolic and mitochondrial fraction at the same size band. In separate cell line, Fad1p-STREP mitochondrial localization is shown to be dependent on an intact mitochondrial membrane potential.  Interestingly, Figure S1 shows the N-term STREP tagged Fad1p is not observed in the mitochondrial fraction but is observed in cytosolic fraction.

Although these results are reasonable, there is no mitochondrial control shown in the western blots. The authors should consider using something like anti-VDAC/Porin on the western membranes to show an appropriate mitochondrial control for the following figures:

  • Figure 2, both panels (a) and (b)
  • Figure 3, panel (a)
  • Figure S1
  • Figure S2 (this figure also does not include the cytosolic control anti-ACTIN; both controls should be included in addition to the specific activity data presented)

Figure 3 shows endogenous Fad1p mitochondrial localization is dependent on carbon source where more Fad1p is mitochondrial localized when grown in non-fermentable carbon sources, and a down regulation is seen when grown on fermentable carbon sources. This is further supported by SDH activity in panel (b). Galactose is an interesting result in that NO Fad1p is observed at all in mitochondria, yet SDH activity is high. The authors do a reasonable job of describing and explaining this result.

R: The mitochondrial fractions used for the analysis of endogenous FAD1p were tested by measuring the specific activity of SDH. In our experiments, a first proof of reliability of SDH activity as a mitochondrial marker is given by the response of the enzyme activity to the different growth conditions, particularly the expected activity repression in the presence of glucose (Figure 3b). The consistency of this marker is even more evident in the mitochondrial sub-fractionation experiment (Figure S2), where the SDH activity significantly decreases in the mitochondrial soluble fraction compared to that measured in the mitochondrial membrane fraction, as expected. In this experiment, the low level of contamination by cytosolic proteins is confirmed by measuring the specific activity of PGI, a standard cytosolic marker, which is almost absent in mitochondrial fractions where the presence of endogenous Fad1p, revealed by immunoblotting, is evident. Furthermore, additional experimental evidence confirming the mitochondrial localisation of Fad1p is given by data shown in Figure 2: the protein is almost undetectable in the presence of the uncoupler FCCP, a typical feature of proteins that are imported into the mitochondria.

Overall, taking also into account that FAD-forming activities have been previously revealed in S. cerevisiae mitochondria (Bafunno V. et al., 2004), we believe that SDH activity measurement is a reliable control to appropriately demonstrate the mitochondrial enrichment with respect to cytosolic proteins and that there is enough experimental evidence supporting Fad1p mitochondrial localisation.

2) The other primary finding of this manuscript is the echoform localization of Fad1p is likely achieved by at least two populations of FAD1 mRNAs with differences in the 3’ UTR length. The authors identify a short 3’ UTR and a long 3’ UTR (Figure 4) and show that total FAD1 mRNA did not depend on carbon source but the amount of long FAD1 mRNA depended on the carbon source (Figure S5).

Based on these data, the authors conclude the 3’ UTR is important for the localization of the mitochondrial echoform. It is not clear though if the FAD1 ORF used in the STREP containing plasmids contains the 3’ UTR, or if the mitochondrial localization observed using these cells is a result of overexpression and the N-terminus of Fad1p. The endogenous detection of Fad1p is much more convincing, but this figure would benefit from additional controls (as mentioned above). In the discussion the authors speculate that Fad1p has a “mitochondrial destination” message in the N-terminus of the protein as it was required for the STREP-Fad1p mitochondrial localization. Further explanation based on the crystal structure of the protein or other relevant information would help clarify this.         

The mechanism regulating localization of natural Fad1p at the transcript level dependent on the carbon source is an ongoing investigation. The authors show that the FAD1 3’ UTR is shared with 3’ UTR for MRP10; the two genes share a M1 mitochondrial localization motif (Figure 5). Mrp10p also lacks a canonical MTS like Fad1p. It was previously demonstrated that MRP10 mRNA interacts with Puf3 via its M1 3’ UTR sequence. Puf3 is an important protein for localizing mRNAs to mitochondrial outer membrane for co-translational insertion into mitochondria. The authors speculate that this mechanism is similar for FAD1 mRNA regulation (Figure 6 – proposed model).

R: We thank the Reviewer for raising this issue. The constructs used for the expression of recombinant Fad1p did not contain both FAD1 UTRs but only the complete ORF (please, see also our response to the Reviewer 2’s comment). This circumstance indicates that the FAD1 mRNA can be translated without being targeted to the outer mitochondrial membrane and the protein is per se capable of entering the mitochondria. In these conditions, however, the mitochondrial localization is not regulated according to the different carbon sources. This fine regulation, on the other hand, would be mediated by the M1 motif and its interaction with specific RNA binding proteins, such as Puf3, located on the outer mitochondrial membrane.

The other crucial determinant for mitochondrial localisation is the presence of a canonical MTS, which is absent at the N-terminus of Fad1p; nevertheless, the Fad1p N-terminal sequence is still important for its mitochondrial import. Indeed, in the crystallographic structure of Fad1p (PDB entry 2wsi), the N-terminal α-1 helix is not in the central core fold of the enzyme, where the enzymatic reaction takes place. This helix, together with α-2 helix in Fad1p are peculiar of Saccharomyces cerevisiae and Candida glabrata proteins, since either they are absent, or they adopt a totally different position in a number of proteins sharing the same core fold (Leulliot N. et al., J Mol Biol. 2010). The N-terminal helix, as well as the contiguous α-2 helix, have indeed amphipathic characteristics, with the basic residues exposed from the same side of the central axis and to the solvent. This may be in agreement with the involvement of this protein moiety in the import process. It is noteworthy that in the human orthologue corresponding regions basic residues are not conserved (see Figure 1). The spatial localisation of the N-terminus part in the human protein is precluded since the produced models largely diverged in the arrangement of this portion (Leone P. et al., Molecules. 2018). Nevertheless, in humans the canonical import signal of the mitochondrial isoenzyme (isoform 1) is located in the N-terminus of the MPTb domain, which precedes the PAPS domain.

As suggested by the Reviewer, we have now added two sentences to the Discussion to better clarify these aspects.

Specific comments regarding the manuscript:

3) Line 48 – Yeasts, as well as fungi, …

Consider changing to “Yeast, as well as other fungi, …” as yeast are a type of fungi and this would clarify the statement.

We have revised the text as recommended.

4) Line 372 – Figure S2 should be changed to Figure S3

It has been done.

Round 2

Reviewer 2 Report

I recommend that the paper be accepted in its current form

Reviewer 3 Report

The authors have sufficiently responded to the points raised regarding the mitochondrial control and the use of SDH activity assay for this, as well as the role of the 3’ UTR in targeting the echoform versus the mitochondrial targeting sequence based on previous structural studies. They have included additional, clarifying text towards this point the manuscript which is appreciated.